# Barriers and enablers of implementing bubble Continuous Positive Airway Pressure (CPAP): Perspectives of health professionals in Malawi

**Alinane Linda Nyondo-Mipando**[1,2]*, **Mai-Lei Woo Kinshella**[3], **Christine Bohne**[4],
**Leticia Chimwemwe Suwedi-Kapesa**[2], **Sangwani Salimu**[2], **Mwai Banda**[2],
**Laura Newberry**[2,5], **Jenala Njirammadzi**[2,5], **Tamanda Hiwa**[2,5], **Brandina Chiwaya**[2],
**Felix Chikoti**[2], **Marianne Vidler**[3], **Queen Dube**[2,6], **Elizabeth Molyneux**[2,6], **Joseph Mfutso-Bengo**[1,2,7], **David M. Goldfarb**[8], **Kondwani Kawaza**[2,5,6], **Hana Mijovic**[9]

1 Department of Health Systems and Policy, School of Public Health and Family Medicine, College of Medicine, University of Malawi, Blantyre, Malawi, 2 College of Medicine, IMCHA Project, Blantyre, Malawi, 3 Department of Obstetrics and Gynaecology, BC Children's and Women's Hospital and University of British Columbia, Vancouver, Canada, 4 Institute for Global Health, NEST 360˚, Rice University, Houston, Texas, United States of America, 5 Department of Pediatrics and Child Health, College of Medicine, University of Malawi, Blantyre, Malawi, 6 Queen Elizabeth Central Hospital, Pediatrics, Blantyre, Malawi, 7 Center of Bioethics for Eastern & Southern Africa (CEBESA), Blantyre, Malawi, 8 Department of Pathology and Laboratory Medicine, BC Children's and Women's Hospital and University of British Columbia, Vancouver, Canada, 9 Department of Pediatrics, BC Children's Hospital and University of British Columbia, Vancouver, Canada

* lmipando@medcol.mw

## Abstract

### Background

Preterm birth complications are the leading cause of neonatal deaths. Malawi has high rates of preterm birth, with 18.1 preterm births per 100 live births. More than 50% of preterm neonates develop respiratory distress which if left untreated, can lead to respiratory failure and death. Term and preterm neonates with respiratory distress can often be effectively managed with Continuous Positive Airway Pressure (CPAP) and this is considered an essential intervention for the management of preterm neonates by the World Health Organization. Bubble CPAP may represent a safe and cost-effective method for delivering CPAP in low-income settings.

### Objective

The study explored the factors that influence the implementation of bubble CPAP among health care professionals in secondary and tertiary hospitals in Malawi.

### Methods

This was a qualitative study conducted in three district hospitals and a tertiary hospital in southern Malawi. We conducted 46 in-depth interviews with nurses, clinicians and clinical supervisors, from June to August 2018. All data were digitally recorded, transcribed verbatim and thematically analyzed.

**Data Availability Statement:** All relevant data are within the manuscript and its Supporting Information files.

**Funding:** ALNM, KK, QD and DG were funded by the Canadian International Development Research Centre (IDRC), Global Affairs Canada (GAC) and the Canadian Institutes for Health Research (CIHR). Project ID is 108030. The funders had no role in study design, data collection and analysis, decision to publish, or preparation of the manuscript.

**Competing interests:** The authors have declared that no competing interests exist.

## Results

Factors that influenced implementation of bubble CPAP occurred in an interconnected manner and included: inadequate healthcare provider training in preparation for use, rigid division of roles and responsibilities among providers, lack of effective communication among providers and between providers and newborn's caregivers, human resource constraints, and inadequate equipment and infrastructure.

## Conclusion

There are provider, caregiver and health system level factors that influence the implementation of bubble CPAP among neonates in Malawian health facilities. Ensuring adequate staffing in the nurseries, combined with ongoing training for providers, team cohesion, improved communication with caregivers, and improved hospital infrastructure would ensure optimal utilization of bubble CPAP and avoid inadvertent harm from inappropriate use.

## Introduction

Neonatal mortality accounts for 46% of all deaths among children under-five years of age worldwide. In Africa alone, it is estimated that approximately one million neonates die in their first four weeks of life, a number that is true in 2017 as it was in 1990 [1]. Complications of prematurity are the leading cause of neonatal deaths [2,3], an important consideration in Malawi, which has the highest rate of preterm births in the world, at 18% of live-births [4].

Many preterm neonates develop respiratory distress and subsequent respiratory failure, contributing to high neonatal mortality [5,6]. Respiratory distress syndrome (RDS) occurs because premature neonates are unable to produce sufficient lung surfactant to maintain adequate breathing [7]. In addition to RDS, pneumonia, pulmonary hemorrhage and sepsis all cause and contribute to respiratory distress in this population [8]. Common causes of respiratory distress in term neonates include but are not limited to pneumonia, meconium aspiration, and transient tachypnea of the newborn [7]. Preterm and term newborns with respiratory distress can often be effectively supported with Continuous Positive Airway Pressure (CPAP), avoiding the need for intubation and mechanical ventilation [6,9,10]. Bubble CPAP is a simple and relatively inexpensive form of CPAP [8,11]. It may represent a safe and cost-effective method for delivering CPAP and reducing neonatal mortality rates in low- and middle-income countries (LMICs) [6,12,13].

Effectiveness of healthcare interventions such as bubble CPAP is critically influenced by the context in which these interventions are being delivered [14]. Limited healthcare infrastructure, including lack of equipment and skilled personnel, may compromise effective use of bubble CPAP in neonatal nurseries [6,12,13,15]. High-quality research studies on effective, safe, and sustainable implementation of bubble CPAP in LMICs are required [6,12].

In high-resource settings, bubble CPAP is set up with tubing, centrally provided medical gases, and oxygen. However, even in the tertiary facilities in Malawi, hospitals do not have access to centrally provided medical gases, and stand-alone bubble CPAP systems that cost over 6000 USD per unit are prohibitively expensive [11]. In 2011, researchers from Rice University developed Pumani, a bubble CPAP system designed for low-resource settings with similar delivery of air pressure and flow to bubble CPAP systems used in high-resource settings.

Pumani is durable, easy to use and repair, is durable [11]. At 800 USD a unit, it is also more affordable than conventional commercial bubble CPAP systems as well.

The Pumani was initially trialed at Queen Elizabeth Central Hospital (QECH) in Malawi in 2012 [5] and from 2012 to 2017, scaled to all 28 central and district hospitals across the country, followed by eight mission hospitals. Along with the installation of the devices, there is training and mentorship of nurses, clinicians and technicians, a regular supply of consumable items needed for its use and quarterly supportive supervisory visits led by Ministry of Health officials [13].

With Malawi's high rate of preterm birth, there is a high need to provide care to newborns with respiratory distress [11]. The objective of this paper is to explore the factors that influence the implementation of Pumani bubble CPAP in Malawi with a focus on what facilitates and what impedes the process. This includes a description of the utilization process and understanding the challenges/barriers and facilitators in its use from the perspectives of health care professionals in four Malawian hospitals where bubble CPAP has been implemented.

Previous studies focused on caregivers experiences with bubble CPAP with minimal attention to healthcare providers' perceptions on factors that influence successful implementation of bubble CPAP at their workplace [16]. Understanding healthcare providers' perspectives on barriers and enablers for implementing bubble CPAP can contribute towards strengthening institutional newborn care in Malawi. Lessons learned from bubble CPAP implementation may be applicable to comprehensive packages of locally appropriate neonatal care technologies, which include but are not limited to bubble CPAP.

## Methods

### Design

We conducted a qualitative study using in-depth interviews to explore the experiences of healthcare professionals that interface with bubble CPAP from their own perspectives. The methodological orientation of grounded theory was employed by the study inductively explore emerging themes [17–19] and in-depth interviews allowed us to investigate the process of using bubble CPAP through rich descriptions of their experiences. The study is reported based on the "Consolidated criteria for reporting qualitative research [20]. Ethics approvals were obtained from the University of Malawi College of Medicine (P.08/15/1783) and the University of British Columbia (H15-01463-A003).

### Research setting

The study was conducted at a tertiary hospital and three secondary level hospitals in the Southern region of Malawi. In consultation with the Malawi Ministry of Health, three districts were chosen because they represented different health management structures available in Malawi as well as different geographical health services zones. District hospital 1 and district hospital 3 are both government hospitals. District hospital 2 is a mission hospital that operates as the district referral hospital in the area. Essential services are provided free of charge at all the facilities.

District level hospitals have Neonatal Care Units staffed by clinicians (medical officers or clinical officers) and nursing staff (registered nurse midwives or nurse midwife technicians). The tertiary hospital has a Neonatal Care Unit as well as several High Dependency Units (HDU) and wards where bubble CPAP is administered to infants and children. These units are staffed by pediatricians, registrars, medical interns, registered nurse midwives and nurse midwife technicians.

Exploration of the barriers and enablers for implementing bubble CPAP is a part of the "Integrating a Neonatal Healthcare Package for Malawi" IMCHA project funded by the Canadian International Development Research Centre (IDRC), Global Affairs Canada (GAC) and the Canadian Institutes for Health Research (CIHR). The project seeks to strengthen institutional newborn care in Malawi through understanding implementation factors for a package of locally appropriate neonatal technologies, including bubble CPAP.

## Recruitment and selection

We drew a purposive sample and included participants involved in health care delivery and/or decision-making for newborn care at the four health facilities. At the district level, district health officers (DHOs), district medical officers (DMOs), district nursing officers, nurses in charge of the pediatric ward, nurses that worked in neonatal units and clinical officers were included. At the tertiary hospital, we included nurses that worked in neonatal units, nurses in charge of the ward, registrars and pediatric consultants. Participants were approached face-to-face or by phone by a member of the research team at both the tertiary and district hospitals and introduced themselves as IMCHA study team members. At the tertiary hospital a registrar approached the possible participants and later the research assistants followed with making appointment with the potential candidate for informed consent procedures and data collection. The registrar did not conduct any interviews. At the district level a program manager who introduced himself as part of the study and was outside the medical hierarchy, connected with participants to book appointments with the potential participants and the research assistants also followed with informed consent procedures and data collection. Based on the number of healthcare professionals that interfaced with bubble CPAP and the limited number of staff available for neonatal care especially at district hospitals, a sample size of 10–15 participants was estimated at each site as being needed to achieve data saturation with a variety of perspectives.

## Interviews

A semi-structured interview guide was developed based on a scoping literature review and preliminary stakeholder consultations. Stakeholders included nurses, physicians, and administrators at the hospitals as well as personnel from the RICE Institute overseeing delivery and utilization of bubble CPAP circuits. The interview guide was piloted with several nurses and physicians at QECH who had experience with bubble CPAP. Pilot data obtained was used to refine phrasing of questions and was not included in analysis. The interview guide was translated into Chichewa, the major local language in Malawi, prior to participant recruitment.

Between June and August 2018, healthcare providers were scheduled for a face-to-face interview of 30–60 minutes at the health facilities in a private setting. A written informed consent form was provided to participants in advance to allow them to fully consider questions about the study and their participation. After completing the consent form in person, interviews were conducted following a semi-structured topic guide on training, initiation, monitoring, differences in opinions, perception and personal experiences and perception on caregiver understanding of bubble CPAP. After the interview, the participant also filled out a demographics form.

Five Malawian researchers, including two certified nurse-midwife-technicians (BC and FC, Diploma in Nursing & Midwifery), two public health specialists (LSK and SS, Masters in Public Health) and a health program manager (MB, Bachelor in Business Administration), were hired as a part of the IMCHA study and underwent a three-day intensive training in qualitative research methods led by ALNM. Three of the researchers conducting the interviews were

female and two were male. Before starting the interview, the researchers introduced themselves and provided the full detail of the study. None of the participants knew the researchers conducting the interview prior to the study. Interviews were conducted mainly in English, which is the language of instruction for health care professions in Malawi, though participants were invited to use the local language of Chichewa if they were more comfortable doing so. The interviews were audio recorded with permission of the participants. Field notes were collected after the interviews. There were no repeat interviews for the study.

## Analysis

Audio files were labelled and provided to qualified and experienced transcribers working with the University of Malawi, College of Medicine to transcribe the recorded interviews verbatim. The completed transcripts were sent to the transcription coordinator (SS), who then reviewed the transcript with the audio to ensure quality and translated any Chichewa terms in the interviews into English where necessary. Transcripts were uploaded to NVivo 12 (QSR International, Melbourne, Australia) as a data management program for qualitative coding. After familiarizing themselves with the transcripts and a review of the field notes with interviewers, two qualitative researchers (MWK, MA in Medical Anthropology, and ALNM, PhD in Health Systems and Policy) developed a codebook (see S1 Appendix) that was both inductive from the data and deductive from the study objectives. Three researchers were involved in the coding with one completing the primary coding of the entire dataset (SS), which was reviewed by two researchers (MWK and ALNM) to verify for soundness and completeness, and add emerging codes. MWK and ALNM then searched and sorted similar codes to group them under overarching themes to plot a thematic map of the process of implementing bubble CPAP.

## Results

We conducted interviews with 46 participants and of these, 27 were females. None of the healthcare providers approached declined to take part in the study and one participant dropped out during an interview due to time constraints. Thirty of the 46 participants were nurses. The median length of service as a healthcare provider was eight years [IQR 3,15] while the median number of years of experience in using the bubble CPAP was three [IQR 1.2,4] (see Table 1).

Five main types of barriers to the implementation of bubble CPAP were identified during all steps of bubble CPAP utilization and included inadequate healthcare provider training in preparation for use, rigid division of roles and responsibilities among healthcare providers, lack of effective communication, human resource constraints and inadequate equipment and infrastructure. Although these factors are grouped into categories, they occurred in an interconnected manner.

### Inadequate healthcare provider training in preparation for use

Participants reported receiving a spectrum of training, from formal to informal. Formal training ranged from an intensive week-long course, mostly in the early stages of implementation of bubble CPAP, to a session at school or a one-day training which was supplemented by on the job observation and practice. Over half (25 of 46 of the health professionals interviewed in the study reported that they were not formally trained on bubble CPAP. Nurses who reported being trained on the job (or simply "orienting") often reported that the training was inadequate and that it mainly focused on monitoring, which included ensuring that the connections to the machines remained intact and checking the baby's vital signs at specified intervals.

**Table 1. Characteristics of study participants.**

| | | Overall | Tertiary Hospital | District Hospital 1 | District Hospital 2 | District Hospital 3 |
|---|---|---|---|---|---|---|
| Total participants | | 46 | 16 | 10 | 10 | 10 |
| Gender | Male | 19 | 3 | 5 | 5 | 6 |
| | Female | 27 | 13 | 5 | 5 | 4 |
| Position | Nurse | 30 | 13 | 6 | 6 | 5 |
| | Clinical Officer | 4 | 0 | 1 | 1 | 2 |
| | District Nursing Officer | 3 | 0 | 1 | 1 | 1 |
| Position Formal training of CPAP | District Medical Officer | 4 | 0 | 1 | 2 | 1 |
| | District Health Officer | 2 | 0 | 1 | 0 | 1 |
| | Pediatric Consultant | 2 | 2 | 0 | 0 | 0 |
| | Pediatric Registrar | 1 | 1 | 0 | 0 | 0 |
| | Yes | 20 | 11 | 4 | 3 | 2 |
| | No | 25 | 5 | 6 | 7 | 7 |
| | Missing | 1 | 0 | 0 | 0 | 1 |

"I can say it [training] wasn't that sufficient because . . .it just prepared me for monitoring only. . . I can say hmm, I am not that comfortable or competent to insert the baby on CPAP because the training that I got was just on job from those who went for training . . . So I can say I am just in between. On a scale of ten, I would say I am at five." District hospital nurse

Echoing the nurse in the quote above, a nurse from the tertiary level facility also shared that those who were trained on the job may not feel competent. Furthermore, when nurses trained on the job oriented others, there may be cascading gaps in knowledge or continuation of potentially harmful practices.

"They [nurses] also need to be trained instead of being trained by us because they do not feel competent enough to do it on their own, so if that person has been taught by someone who is not competent enough to put the baby on CPAP then it becomes a problem because he/she will be doing things in a wrong way." Tertiary hospital nurse

The process of weaning the baby off bubble CPAP was identified as a gap in knowledge secondary to inadequate training. For example,

"I get confused as to how I should go about it as to whether I have to. . .reduce the oxygen levels or reduce the water levels together at the very same time or not or am I supposed to do it one by one. . .so these things always confuses me" Tertiary hospital nurse

She continued to state that not only nurses were confused but also the clinicians,

"I think that our doctors are not well trained more especially the registrars and the consultants because they do not understand the process of weaning a child as well, so weaning a child is supposed to be 5cm water level, so there was this other time when the doctor ordered that we should reduce from 5cm to 4cm then 3 and eventually we reached at 2cm, so we had some disagreements in the process, so I believe that sometimes our doctors have little information on some of these issues for example the issue of weaning a child process" Tertiary hospital nurse

Participants shared that delays in weaning may cause nasal complications from over-staying on bubble CPAP as well as lead to further delays in initiation as CPAP system is occupied. They wished for comprehensive and regular training that provided consistent information on how the CPAP system worked, indications/contra-indications, process of initiation, monitoring and weaning, troubleshooting as well as hands-on skills practice was considered to facilitate its use. Learners also appreciated multimedia platforms such as learning through videos.

> "We had done a formal training; it was a good training. The facilitators were good and the tutorials. . . captured what was intended and useful. . ..The people who presented the contents showed that they really had the knowledge about what they were to present and also the mode of delivery of the contents were in line with adult learning [with] videos, hands on. . .." District hospital nurse

## Rigid division of roles and responsibilities among healthcare providers

There were a variety of opinions reflected in clinical practice, as the healthcare providers described clinical situations where there was a disagreement on when to initiate a newborn on bubble CPAP. Medical hierarchy, including the roles and responsibilities of different cadres, sometimes complicated decision-making. While nurses were often the recipients of training as direct users of the system, clinicians held higher authority in decision-making. For example, a nurse from the central hospital narrated a scenario, citing difficulties especially with new clinicians:

> "We said that the child was not supposed to be put on CPAP because of the cardiac problem. . . so we discussed [it] with doctors but they insisted to put the baby on CPAP, so the baby was put on CPAP and the condition was still worsening. . . [and] eventually the baby died. . .. It becomes so hard for us to argue with them since they are doctors and they are above us because the moment you say we do not do these things like that, they think that we are underrating them, so these doctors need to be trained more. . . for example, the registrars and consultants" Tertiary hospital nurse

Even when nurses were trained and encouraged to make the decision to initiate bubble CPAP, some nurses reported reluctance for fear that the clinician would question their decision afterwards.

> "They (nurses) were briefed, yes, yes but they still don't want to commence. May be they are trying to, they still feel that I can't do this. I still have to wait for the clinician, feel probably [that] they will question me on why I did this." District hospital nurse

## Lack of effective communication

Communication gaps between nurses and clinicians resulted in delays on starting the baby on bubble CPAP. Below is an example where a lack of communication led to a delay in initiation:

> "So what some clinician do. . . they just order, leaving the file there, and they don't even say. . .go and put the baby on CPAP. That one time. . . the clinician ordered CPAP on the baby without telling us and he went. Maybe he forgot, I don't know, but he didn't tell us (nurses). We were also busy and it took us time for us to see the orders. It was morning around 8:00 and by the time we discovered that the baby was supposed to be on CPAP, it was around 4:00pm." District hospital nurse

The nurse's quote above reveals how poor communication combined with rigid division of responsibilities between clinicians prescribing and nurses completing the task can exacerbate delays in initiation. Additionally, the quote also highlights that nurses often had heavy burdens of care.

In addition to communication challenges among medical staff, there were also communication challenges between healthcare providers and the newborn's caregivers. Study participants reported that caregivers sometimes had fears that the many tubes interfered with breathing and that oxygen therapy was associated with death–a perception that may have been influenced by the lack of clear, effective communication between providers and caregivers. In the tertiary hospital nursery, visiting hours were limited and some providers were concerned about initiating bubble CPAP before they were able to obtain consent from caregivers. Clinicians and nurses interviewed spoke about the need to counsel caregivers and get their consent before initiating CPAP. In reality, time constraints of healthcare providers and difficulties explaining CPAP in lay language at times lead to inadequate explanations for caregivers:

> "The first thing is fear. They are just afraid that these are machines. . . ..most of health workers have that perception that if they understand something in the hospital then the mother should obviously also understand, and because of that perception we don't take much time explaining to the mother because we assume that they already understand just because we understand. . ." District health officer

Effective communication and cohesion between healthcare providers as well as between providers and caregivers were reported to support use of bubble CPAP. The quotes below illustrate how effective communication can improve cohesion of the medical team to work together on decision-making, gathering the supplies needed, and enabling caregivers to understand the procedure.

> "We always inform our ward in-charge and this person is supposed to report the issue to our matron and this matron makes sure that she gets or find these materials so that we should be able to use them and again, we also inform the doctor on duty so that he/she should also know about these issues so that together we should look into the matter and see how we can help each other on the issue" Tertiary hospital nurse

> "You have to understand, you know when a baby is sick, a mum is also sick. So you have to be walking together with the mum. . . So number one is your relationship with the guardian. Because that relationship will also go together with explaining everything that you want to do so that the guardian can understand. Again, what can make them not accept is how you explain." District medical officer

## Human resource constraints

Staffing shortages, especially at night when only one nurse was on duty may also contribute to delayed initiation until they completed other tasks. For example, one nurse recalled,

> "The nurse on duty was busy. . .the nurse was alone, and sees this child needs CPAP. . .and had the woman who had severe bleeding. . .so to her side. . .eh. . ..had to save the life of the mother who was severely bleeding" District hospital nurse

Staffing shortages further affected monitoring of patients on bubble CPAP, including checking over the machine, connections to the baby, as well as vital signs. Study participants

reported that monitoring was ideally supposed to be done in the first 15 minutes to assess the infants breathing rate, then an hour after initiation and then every four to six hours. While this is the ideal, it was not often followed because of nurse shortages especially at night and other responsibilities. For example:

". . .so because of workload sometimes we were doing the monitoring sometimes we were failing to do the monitoring as planned because of staffing issues. . .with the staffing sometimes we would miss them. . .So for some yes they would miss them for the whole 24 hours, some would miss them for twelve hours" District hospital nurse

The practice of rotating nurses through the different wards of the hospital created shortage of bubble CPAP formally trained staff since the rotations disregard the trainings one received and meant that those who received formal training did not stay in the nursery.

"Because most of them [nurses] who were trained in CPAP are not involved . . ..they were trained but today they are in other wards. . ..people should not just be initiating CPAP on a baby by just using the manual. . ..I think that that is not fair" District hospital nurse

Staffing shortages and priorities of service responsibilities made it a challenge for nurses and district health supervisors to have the time away from their duties for formal training.

"Looking at the way this ward works, it is always busy so you cannot say let me have my own training leaving some other children who are sick here and you cannot leave him/her just like that without attending to the baby simply because you were not trained" Tertiary hospital nurse

To cope with the shortage of staffing, caregivers were highlighted as a resource for monitoring. As one district medical officer said,

"In many cases, as you know, we might not have enough hands on the ground so sometimes the mother also helps in monitoring the baby. . .So you tell the mother that this will help the baby breathe better and for us to make sure that the machine is working, we have to see the bubbles in the bottle. . . If it stops bubbling, then. . .tell the one staff on duty" District medical officer

Particularly in district hospitals where nurses may be responsible for different areas around labor, delivery and postnatal care, caregivers who are especially attentive to the wellbeing of their baby could help support basic monitoring. However, monitoring by clinical staff was felt to be still required for vital signs.

Nurses' spoke of how having nurses that are specifically dedicated or assigned to the nursery would help with regular monitoring. However, even when there were nurses assigned to the nursery, realities of staffing shortages meant a lack of separation for nursery, postnatal and labour ward nursing staff in district hospitals.

"In fact, we advocated that the nursery should be taken as a ward on its own. . .but because of the shortages, we have just combined the post-natal and the nursery" District hospital nurse

### Inadequate equipment and infrastructure

Human resource limitations were further exacerbated by constraints around medical equipment and infrastructure, especially lack of electricity and accessories that require appropriate sizing like nasal prongs:

> "We do face some small challenges when using CPAP since we have the prongs which enters into the nostrils so when these prongs are too big for the baby it makes the nose of the baby to go up causing the nose to disfigure in the process" Tertiary hospital nurse

> "Whenever you put CPAP you make sure that electricity is there so electricity is also a problem. It's almost daily, it's daily to have blackouts and generator to be ignited or put on, it requires management, to be asked or to be authorised by management. So it's quite challenging you find that the baby needs CPAP and there in no power." District hospital clinical officer

## Discussion

Factors affecting bubble CPAP implementation were interconnected and the barriers included inadequate healthcare provider training in preparation for use, rigid division of roles and responsibilities among healthcare providers, lack of effective communication, human resource constraints, and inadequate equipment and infrastructure. Factors that facilitated implementation of bubble CPAP were comprehensive training, team cohesion, dedicated nurses for the nursery and consistent availability of electricity and equipment.

Previous work in sub-Saharan Africa largely examined barriers and enablers as issues that emerged from efficacy and safety studies [5,13,21,22]. Training that enabled effective use was comprehensive, included hands-on practice, troubleshooting problems and weaning, as well as training both clinicians and nurses so there is a common understanding and better team cohesion. Team cohesion was also related to the need to strengthen communication channels. Lastly, both human and infrastructural resources were fundamental to effective bubble CPAP use including dedicated neonatal nurses and consistent availability of electricity and equipment.

Training may occur on-the-job through observation and mentorship as described by many of the health professionals in our study, which has been found to be more effective when reinforced with use of standardized tools that illustrate and engage the learner [23]. While our research findings agree with previous recommendations for regular training, mentorship and investment in nursing staff [22,24,25], our study suggests that efforts must go deeper to understand the health system issues around staffing of nursery wards, empowerment of nurses and developing medical team cohesion. Currently, a guided mentorship by experienced clinical staff has been introduced in Malawi district hospitals in order to help with identifying and treating eligible newborns [13,26]. Investment into training and mentorship for nursing staff must also be taken into context of the overall small pool of nurses assigned to the nursery, the common practice of rotating staff between wards where the mentors and trained nurses may not remain in the nursery and medical hierarchy and power dynamics around who owns the decision to initiate. Improvements for health care delivery require inter-professional education on clinical competencies and a deliberate effort of empowering nurses in clinical decision-making supported by policies to resolve workplace issues [27].

Our findings that rigid roles and responsibilities among providers result in nurses feeling incompetent to make clinical decisions is similar to studies highlighting the importance of

competency confidence in nurses' clinical decision-making [27] and conversely, how hierarchical organizational structures can inhibit clinical decision-making by nurses [28]. Rigid roles and responsibilities leads to delays in initiation of bubble CPAP and time is a key factor in the outcome of the neonate in a critical situation. Kawaza and colleagues showed that initiation delays were associated with longer stays in the hospital and twice as long on treatment in comparison to those who received bubble CPAP earlier [5]. A Kenyan study reported training-of-trainers concept with both nurses and clinicians together was effective and demonstrated that nursing staff were able to initiate bubble CPAP [29]. Providers in our study highlighted how rigid divisions of roles and responsibilities and lack of training for those with decision-making authority negatively affected initiation of bubble CPAP.

Furthermore, poor communication between the clinical team and between the clinical team and caregivers contributed to delays between the decision to initiate and initiating the baby. Patient care has been found to be compromised when nurses and doctors work in isolation to each other with ineffective communication with each other, especially within hierarchical authority structures where clinicians are seen as the primary clinical decision-maker for a patient [30,31]. A "conductor-less orchestral model" where improved communication and recognition of the efforts of all healthcare workers' contribution to patient care [30] and interdisciplinary training to promote team-building between different professionals [32] may help mitigate the reification of the hierarchical structures that compromise quality care.

Our findings on shortage of neonatal staff are in line with previous research on bubble CPAP in sub-Saharan Africa where understaffed neonatal units and high turnover of nurses and doctors limited capacity for care [9,24,25,33,34]. Currently, Malawian government hospitals have 29% vacancies for medical officers, 63% for clinical officers, 66% for registered nurse-midwives and 60% for nurse-midwife-technicians, which affects the provisioning of neonatal services [35]. In addition to compromising capacity for care, shortages of staff and staff rotations between wards negatively affected the ability for staff to go for training and to retain trained staff in the nursery. As Malawi is advancing the training of health personnel, there is need to consider specialty training of neonatal nurses who stay in nurseries without rotation to other wards. Our findings on constraints resulting from inadequate human resources resonates with a previous qualitative study on health care worker perspectives on bubble CPAP in India where although CPAP was accepted by health care workers, its use was impeded by shortages of staff, along with equipment [36].

In addition to a lack of nurses dedicated to the nursery, there are few clinicians at the district hospitals. Particularly in the rural district hospitals, written instructions and checklists may boost nurse's confidence and strengthen task-sharing of initiation and management of bubble CPAP among neonates [24]. Nurses in the district hospitals could be empowered to initiate bubble CPAP as has been done in Kenya [29,34]. Mobile health platforms may be an effective approach to support the healthcare worker training on-the-job and consistency of training [37,38], two key issues raised in our research. Two innovative studies in Nigeria [39] and South Africa [40] has highlighted the potential for using a WhatsApp platform for teaching, supervising and supporting students in their integration of theory with clinical practice. There is potential for a similar program to support health care workers in the training and mentorship for bubble CPAP.

The Pumani bubble CPAP system does not have a built in humidifier, which required regular nasal saline drops [5]. While nasal complications were sometimes mentioned, they were more frequently highlighted as a problem of inappropriately sized nasal prongs and the baby overstaying on bubble CPAP. However, infrequent mentions of the nasal saline drops may not mean that they are not an issue. Delays in weaning increased the risk of complications and added to human resource and equipment needs that in turn contributed to delays in initiation

for other babies. Barriers earlier in the utilization process, including human and material resource constraints leading to delays or lack of initiation, may mask some potential challenges around monitoring and weaning.

Health professionals shared that they understood that using bubble CPAP required a number of items, from tubing to appropriate sized nasal prongs, an oxygen source, reliable electricity and others, and that babies on bubble CPAP needed to be monitored closely to minimize inadvertent harm. This meant that health professionals hesitated to initiate when there were gaps in material and human resources.

While this paper considered implementation factors for bubble CPAP utilization in resource-constrained settings found in Malawian hospitals more generally, an intervention implemented in a tertiary hospital will not involve the same process as in a secondary level facility. Consequently, future research is planned to tease out implementation factors in tertiary versus secondary level district hospitals to better understand the best ways to support healthcare workers in the different contexts. Future research should also explore the implementation of mHealth approaches to support training and continued mentoring and coaching on bubble CPAP. Furthermore, research should consider inclusion of decision makers and other cadres of staff with more influence on what interventions get implemented within the Malawi Health System.

## Strengths and limitations

Our study adds to the existing literature by closely examining the factors that influence implementation of bubble CPAP from healthcare professionals' perspectives. Furthermore, our paper navigates through the process of bubble CPAP implementation from training, decision-making to initiate, putting the baby on bubble CPAP, monitoring and weaning. Although our study followed a qualitative approach, which may limit generalisations of the results, our findings offer concepts that can be taken onboard when implementing bubble CPAP in other hospitals. It is important to emphasize that preterm neonates often have complex medical needs that are not limited to respiratory distress. Comprehensive newborn care is therefore required to achieve tangible improvements in neonatal mortality. While we did not explore other neonatal care interventions in our bubble CPAP interviews, we are cognizant that CPAP should be a component of a larger package of newborn care.

Researchers' background and motivations inevitably influence the researcher-study participant dynamics and therefore participants' responses. Healthcare providers participating in our study were aware that all the researchers were part of the IMCHA program, which supports implementation of neonatal technologies, and works in close partnership with RICE Institute, which supports delivery and utilization of bubble CPAP in Malawi. While the researchers made it clear that the objective of this research study was not to evaluate healthcare providers' performance or to provide additional hospital resources, we acknowledge that respondent's desirability bias could not be avoided. A similar study conducted by an independent research team may have elicited additional barriers to bubble CPAP implementation that were omitted during our interviews.

Researcher conducting the interviews came from clinical (nurses) and non-clinical (public health) backgrounds. Upon reflecting on the interviews, nurses noted that they had to reemphasize their role as researchers rather than clinicians with study participants and ask them to explain concepts that may only be familiar to nurses in lay language.

Finally, our study focused on the perspectives and experiences of frontline healthcare providers and district level health management. These findings will have to be triangulated with perspectives from healthcare policy decision makers, such as government officials, who are

able to speak about barriers and facilitators to bubble CPAP implementation from a system level perspective.

## Conclusion

Implementation research on barriers and facilitators to the use of novel interventions in low-resource settings with high neonatal mortality is important for the effective scaling of essential technologies. Our research on implementation factors for the use of bubble CPAP for neonates in Malawian hospitals from the perspective of healthcare professionals revealed an interconnection of provider, caregiver and health system level factors that contribute to delays in its use but also highlights potential areas where the implementation of bubble CPAP can be strengthened for more effective use.

Ensuring adequate staffing in the nurseries, combined with ongoing training for providers, team cohesion, improved communication with caregivers, and improved hospital infrastructure would ensure optimal utilization of bubble CPAP and avoid inadvertent harm from inappropriate use. Lessons learned from bubble CPAP implementation may be applicable to comprehensive packages of locally appropriate neonatal care technologies. Our study was conducted in hospitals with varying authority structures, ownership, and across different cadres of healthcare providers, therefore offering a holistic view of the implementation of bubble CPAP.

## Supporting information

**S1 Appendix. Codebook.**
(DOCX)

**S1 File. Healthcare workers experiences with bubble CPAP in neonatal nurseries.**
(DOCX)

## Acknowledgments

We are grateful to all the study participants that participated in the study and the nurses who helped in data collection. We are thankful for the institutional support from the Hospitals for allowing us to conduct the study in their facilities and Rice University for their support. Dr. Nyondo-Mipando is supported by Malawi HIV Implementation Research Scientist Training program (Fogarty: D43 TW010060).

## Author Contributions

**Conceptualization:** Alinane Linda Nyondo-Mipando, Laura Newberry, Jenala Njirammadzi, David M. Goldfarb, Kondwani Kawaza.

**Formal analysis:** Alinane Linda Nyondo-Mipando, Mai-Lei Woo Kinshella, Sangwani Salimu, David M. Goldfarb, Kondwani Kawaza.

**Funding acquisition:** Queen Dube, Kondwani Kawaza.

**Investigation:** Alinane Linda Nyondo-Mipando, Christine Bohne, Leticia Chimwemwe Suwedi-Kapesa, Sangwani Salimu, Mwai Banda, Tamanda Hiwa, Brandina Chiwaya, Felix Chikoti.

**Methodology:** Alinane Linda Nyondo-Mipando, Hana Mijovic.

**Project administration:** Mwai Banda.

**Supervision:** Alinane Linda Nyondo-Mipando, Mwai Banda, Marianne Vidler, Queen Dube, Elizabeth Molyneux, Joseph Mfutso-Bengo, David M. Goldfarb, Kondwani Kawaza, Hana Mijovic.

**Visualization:** Mai-Lei Woo Kinshella.

**Writing – original draft:** Alinane Linda Nyondo-Mipando, Mai-Lei Woo Kinshella, Christine Bohne.

**Writing – review & editing:** Alinane Linda Nyondo-Mipando, Mai-Lei Woo Kinshella, Christine Bohne, Leticia Chimwemwe Suwedi-Kapesa, Sangwani Salimu, Mwai Banda, Laura Newberry, Jenala Njirammadzi, Tamanda Hiwa, Brandina Chiwaya, Felix Chikoti, Marianne Vidler, Queen Dube, Elizabeth Molyneux, Joseph Mfutso-Bengo, David M. Goldfarb, Kondwani Kawaza, Hana Mijovic.

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
