## [Decision Letter · Decision Letter 0]

17 Sep 2019

PONE-D-19-22572

Barriers and enablers of implementing bubble Continuous Positive Airway Pressure (CPAP): Perspectives of health professionals in Malawi

PLOS ONE

Dear Dr Nyondo-Mipando,

Thank you for submitting your manuscript to PLOS ONE. After careful consideration, we feel that it has merit but does not fully meet PLOS ONE’s publication criteria as it currently stands. Therefore, we invite you to submit a revised version of the manuscript that addresses the points raised during the review process.

Please see comments from both reviewers below.

We would appreciate receiving your revised manuscript by Nov 01 2019 11:59PM. To enhance the reproducibility of your results, we recommend that if applicable you deposit your laboratory protocols in protocols.io, where a protocol can be assigned its own identifier (DOI) such that it can be cited independently in the future. For instructions see: http://journals.plos.org/plosone/s/submission-guidelines#loc-laboratory-protocols

We look forward to receiving your revised manuscript.

Kind regards,

Charles A. Ameh, PhD, MPH, FWACS (OBGYN), FRCOG

Academic Editor

PLOS ONE

Journal Requirements:

2. Please include a copy of the interview guide used in the study, in both the original language and English, as Supporting Information, or include a citation if it has been published previously.

Additional Editor Comments (if provided):

Thanks for submission to PloS one, both reviewers have made comments that need to be addressed before your article can be considered for publication.

Reviewers' comments:

Reviewer's Responses to Questions

**Comments to the Author**

1. Is the manuscript technically sound, and do the data support the conclusions?

Reviewer #1: Yes

Reviewer #2: Yes

2. Has the statistical analysis been performed appropriately and rigorously? 

Reviewer #1: N/A

Reviewer #2: N/A

3. Have the authors made all data underlying the findings in their manuscript fully available?

Reviewer #1: Yes

Reviewer #2: No

4. Is the manuscript presented in an intelligible fashion and written in standard English?

Reviewer #1: No

Reviewer #2: Yes

5. Review Comments to the Author

Reviewer #1: This is a very interesting paper, exploring the views of healthcare workers from a tertiary centre and district hospitals in Malawi, about using CPAP in neonates in the current Malawian context of health care delivery. There is an increasing interest in using CPAP in LMICs. However, many studies describe how CPAP is effectively used, but few studies explore in-depth the multifaceted aspects of using this technology in LMICS. This paper contributes importantly to understanding in-depth the issues of implementing and using CPAP in a LMIC setting.

Most of my comments are minor. There is only one major comment which is to change the structure of presenting the different themes identified in this study (see my comments about the Discussion section below). This would improve the clarity of the major themes, make the paper easier to read, and allow identifying the main messages of the paper more rapidly.

ABSTRACT

Background:

The way the background is written might be interpreted that only preterm babies would benefit from CPAP. Although it is true that they are the main group of neonates to benefit from it, term babies can also present respiratory distress (respiratory distress in general, not RDS) for other reasons than prematurity. Consider adding this consideration.

Conclusion:

Line 51: Patient factors are mentioned here, but actually in the main paper it is the perceptions of caregivers that are reported, which are not patient factors as such. To me, patients factors would be severity of disease, gestational age, etc..

Line 53: Operations of health system is pretty vague. Consider fleshing it out a bit.

MAIN PAPER

Introduction:

As mentioned before, CPAP is also useful for non-preterm babies.

Line 76. I suggest replacing “Wall air” with “centrally provided medical gases”? or something like that.

The introduction needs to say a bit more about what is the knowledge gap and why this study is needed. Were there issues with the implementation of the Pumani CPAP that made you want to explore in depth the views of healthcare workers? Information in lines 111-116 p 4 could be used in the introduction to provide more information about why the study has been conducted.

Methods:

Line 99 through not though

What was the methodological orientation (presumably grounded theory?)

Lines 104-105 why these DH in particular and not any other the 28 other centres presented in introduction. Are these three centre similar or different compared to the other centres?

Recruitment:

Page 5 line 130 “a” scoping review not “the” scoping review

Line 138-142 is too long need to be split in a couple of sentences.

Page 6 line p160 grounded theory should come earlier in the methods section

Line 161 Nvivo was use for data management purposes, not for the analysis as such (which was conducted by the authors)

How many people who were invited refused to take part and why?

Results:

The findings (note that implementation and use of CPAP is not the same) are not presented in a consistent way in the Abstract and in the Results section:

Abstract: Results section/ figure 1:

Factors that influence implementation of CPAP Factors that influence utilization of CPAP/contributes to delays

1. lack of confidence due to inadequate training 1. Training in preparation for use

2. hierarchy of decision-making in combination with poor communication among healthcare providers; 2. Decision making to initiate CPAP and initiation

3. human resource constraints 3. Monitoring the neonate on bubble cpap

4. and gaps in infrastructure and supplies 4. Weaning the neonate off bubble cpap

In addition, there are overlaps across the different factors presented in Results and figure 1. For example “human resource constraints” delay initiation onto bubble cpap, but also affects monitoring while on bubble cpap. “Inadequate training” delays preparation of staff, and also have a negative impact on weaning.

The paper would be clearer if the identified themes are internally coherent, consistent, and distinctive. I Think the first column of table 2 actually provides a list of internally coherent, consistent, and distinctive themes:

1. Inadequate training is an issue (here you can describe both the problem of lack of training in preparation for use, on weaning process, and on any other aspect of CPAP use)

2. Lack of effective communication is an issue (here you can have two sub themes:

a. Lack of communication between health care providers such as between the nurses and clinicians

b. Lack of communication between healthcare workers and parents)

3. Parents fear bubble CPAP machine

4. There are human resources constraints (here you can present the lack of staff, and the rotation to other wards issue)

5. Rigid division of roles and responsibilities

6. Lack of equipment and infrastructure

The second column “enablers” of table 2 mirrors colum1; it doesn’t really add anything.

Consider structuring the Results section around the 6 themes listed above.

Consider a short sentence to introduce the themes identified in the study at the beginning of the results section. Readers need to be prepared to what will follow. For example: “Six themes emerged from the analysis: 1) Inadequate training…, 2) Lack of effective communication…, 3) Parents fear bubble CPAP, 4)…, etc.” It would be much easier to read, and to understand the main messages of the paper.

Then, you should use the same structure for the results section in the Abstract, and the first paragraph (principal findings) in the the Discussion section.

Line 186: there are inconsistencies in the order of presenting broad categories (health system, providers, caregivers) throughout the manuscript.

Abstract Results

Line 51: There are personal, patient and health system level factors Line 186: at the health system, provider and caregiver level

Lines 195 percentages are usually not used in a qualitative paper.

Line 198 provide more details about what monitoring means

Discussion

The discussion should start with main findings.

There should be a “strength and limitations” section in the discussion. P14, Lines 358-360 is a strength of the study

You should present the themes in the same order than in the Abstract and the Results section

The following points in the Discussion are not presented beforehand in the Results section:

• Line 429-430 “However, health professionals in our study rarely discussed the need for nasal saline drops as a challenge”.

• Line 442-44 “To avoid the increased burden of monitoring a baby on bubble CPAP, some nurses shared that they left the baby on nasal oxygen which reportedly required less issues to monitor.”

Suggestions: either present them in the results section, or remove them.

Reviewer #2: This was a very well-written manuscript from a study that was well-designed and conducted. The details of the study are clearly stated, and justification of methods are clearly outlined. The data analysis was rigorously done. The tables and figures are well-presented and described in the text.

Discussion:

Line 395 “Bubble” remove capital.

Although alluded to in the last paragraph of this section, it would be useful for the authors to provide more details on their reflections of the limitations of this study and how they have attempted to mitigate them.

Well done to the team.

6. PLOS authors have the option to publish the peer review history of their article (what does this mean?). If published, this will include your full peer review and any attached files.

Reviewer #1: Yes: Juan Emmanuel Dewez

Reviewer #2: No

---

## [Author Response · Author response to Decision Letter 0]

30 Oct 2019

PONE-D-19-22572

Barriers and enablers of implementing bubble Continuous Positive Airway Pressure (CPAP): Perspectives of health professionals in Malawi

PLOS ONE

On behalf of the research team, I thank you for your thorough and constructive review of our manuscript. Please find below our responses to the queries raised. Please note that following the revisions made to the manuscript, we have deleted our Figure 1 as it was repetitive and was not adding any more value. Note that the stated dates in this letter of response are as they are in the clean version of the manuscript. 

5. Review Comments to the Author

Reviewer #1: This is a very interesting paper, exploring the views of healthcare workers from a tertiary centre and district hospitals in Malawi, about using CPAP in neonates in the current Malawian context of health care delivery. There is an increasing interest in using CPAP in LMICs. However, many studies describe how CPAP is effectively used, but few studies explore in-depth the multifaceted aspects of using this technology in LMICS. This paper contributes importantly to understanding in-depth the issues of implementing and using CPAP in a LMIC setting.

Most of my comments are minor. There is only one major comment which is to change the structure of presenting the different themes identified in this study (see my comments about the Discussion section below). This would improve the clarity of the major themes, make the paper easier to read, and allow identifying the main messages of the paper more rapidly.

Response: We appreciate the comment and have revised the presentation of our results as suggested and they are clarified under specific comments where they were highlighted. Refer to lines 192 to 390.

ABSTRACT

Background:

The way the background is written might be interpreted that only preterm babies would benefit from CPAP. Although it is true that they are the main group of neonates to benefit from it, term babies can also present respiratory distress (respiratory distress in general, not RDS) for other reasons than prematurity. Consider adding this consideration.

Response: We have revised the background to include the use of CPAP among term babies as follows: Term and preterm neonates with respiratory distress can often be effectively managed with Continuous Positive Airway Pressure (CPAP) and this is considered an essential intervention for the management of premature neonates by the World Health Organization. Refer to lines 33 to 35.

Conclusion:

Line 51: Patient factors are mentioned here, but actually in the main paper it is the perceptions of caregivers that are reported, which are not patient factors as such. To me, patients factors would be severity of disease, gestational age, etc..

Response: We have revised this statement and it reads as follows: There are provider, caregiver and health system level factors that influence the implementation of bubble CPAP among neonates in Malawian health facilities. Refer to lines 49 to 50.

Line 53: Operations of health system is pretty vague. Consider fleshing it out a bit.

Response: We have clarified this sentence and it reads as follows:

Ensuring adequate staffing in the nurseries, combined with ongoing training for providers, team cohesion, improved communication with caregivers, and improved hospital infrastructure would ensure optimal utilization of bubble CPAP and avoid inadvertent harm from inappropriate use. Refer to lines 50 to 53.

MAIN PAPER

Introduction:

As mentioned before, CPAP is also useful for non-preterm babies.

Response: We have included the information that CPAP is also used in non-preterm babies and it now reads as follows:

Many preterm neonates develop respiratory distress and subsequent respiratory failure, contributing to high neonatal mortality (5,6). Respiratory distress syndrome (RDS) occurs because premature neonates are unable to produce sufficient lung surfactant to maintain adequate breathing. In addition to RDS, pneumonia, pulmonary hemorrhage and sepsis all cause and contribute to respiratory distress in this population. Common causes of respiratory distress in term neonates include but are not limited to pneumonia, meconium aspiration, and transient tachypnea of the newborn. Preterm and term neonates with respiratory distress can often be effectively supported with Continuous Positive Airway Pressure (CPAP), avoiding the need for intubation and mechanical ventilation. Refer to lines 63 to 71.

Line 76. I suggest replacing “Wall air” with “centrally provided medical gases”? or something like that.

Response: We have revised “wall air” with “centrally provided medical gases”. Refer to line 80 and 82.

The introduction needs to say a bit more about what is the knowledge gap and why this study is needed. Were there issues with the implementation of the Pumani CPAP that made you want to explore in depth the views of healthcare workers? Information in lines 111-116 p 4 could be used in the introduction to provide more information about why the study has been conducted.

Responses: We have highlighted the knowledge gap and it is as follows:

Previous studies focused on caregivers experiences with bubble CPAP with minimal attention to healthcare providers’ perceptions on factors that influence successful implementation of bubble CPAP at their workplace (16).Understanding healthcare providers’ perspectives on barriers and enablers for implementing bubble CPAP can contribute towards strengthening institutional newborn care in Malawi. Lessons learned from bubble CPAP implementation may be applicable to comprehensive packages of locally appropriate neonatal care technologies, which include but are not limited to bubble CPAP. Refer to lines 101 to 107.

Methods:

Line 99 through not though

Response: This has been corrected and it is in line 115.

What was the methodological orientation (presumably grounded theory?)

Response: We have specified earlier within the methods section that the methodological approach was a grounded theory. Refer to line 113.

Lines 104-105 why these DH in particular and not any other the 28 other centres presented in introduction. Are these three centre similar or different compared to the other centres?

Response: We have added more information on the Districts however we have limited the information for ethical reasons to avoid unintended disclosure of the specific facility. We have added the following statement:

Research Setting: In consultation with the Malawi Ministry of Health, three districts were chosen because they represented different health management structures available in Malawi as well as different geographical health services zones. District hospital 1 and district hospital 3 are both government hospitals. District hospital 2 is a mission hospital that operates as the district referral hospital in the area. Essential services are provided free of charge at all the facilities. Refer to lines 121 to 126

Page 5 line 130 “a” scoping review not “the” scoping review

Response: This has been revised and is reflected in line 152

Line 138-142 is too long need to be split in a couple of sentences.

Response: The sentence has been split.

Page 6 line p160 grounded theory should come earlier in the methods section

Response: We have brought the information regarding the methodological approach to the study upfront under the design section as follows: The methodological orientation of grounded theory was employed by the study inductively explore emerging themes. Refer to line 113.

Line 161 Nvivo was use for data management purposes, not for the analysis as such (which was conducted by the authors)

Response: We have revised this to reflect that NVIVO was used for management and not analysis and it reads as follows:

Transcripts were uploaded to NVivo 12 (QSR International, Melbourne, Australia) as a data management program for qualitative coding. Refer to lines 182-183.

How many people who were invited refused to take part and why?

Results: None of the health workers approached declined to take part in the study and only one withdrew during interviews secondary to time constraints. 

The findings (note that implementation and use of CPAP is not the same) are not presented in a consistent way in the Abstract and in the Results section:

Abstract: Results section/ figure 1:

Factors that influence implementation of CPAP Factors that influence utilization of CPAP/contributes to delays

1. lack of confidence due to inadequate training 1. Training in preparation for use

2. hierarchy of decision-making in combination with poor communication among healthcare providers; 2. Decision making to initiate CPAP and initiation

3. human resource constraints 3. Monitoring the neonate on bubble cpap

4. and gaps in infrastructure and supplies 4. Weaning the neonate off bubble cpap

In addition, there are overlaps across the different factors presented in Results and figure 1. For example “human resource constraints” delay initiation onto bubble cpap, but also affects monitoring while on bubble cpap. “Inadequate training” delays preparation of staff, and also have a negative impact on weaning.

The paper would be clearer if the identified themes are internally coherent, consistent, and distinctive. I Think the first column of table 2 actually provides a list of internally coherent, consistent, and distinctive themes:

1. Inadequate training is an issue (here you can describe both the problem of lack of training in preparation for use, on weaning process, and on any other aspect of CPAP use)

2. Lack of effective communication is an issue (here you can have two sub themes:

a. Lack of communication between health care providers such as between the nurses and clinicians

b. Lack of communication between healthcare workers and parents)

3. Parents fear bubble CPAP machine

4. There are human resources constraints (here you can present the lack of staff, and the rotation to other wards issue)

5. Rigid division of roles and responsibilities

6. Lack of equipment and infrastructure

The second column “enablers” of table 2 mirrors colum1; it doesn’t really add anything.

Consider structuring the Results section around the 6 themes listed above.

Consider a short sentence to introduce the themes identified in the study at the beginning of the results section. Readers need to be prepared to what will follow. For example: “Six themes emerged from the analysis: 1) Inadequate training…, 2) Lack of effective communication…, 3) Parents fear bubble CPAP, 4)…, etc.” It would be much easier to read, and to understand the main messages of the paper.

Then, you should use the same structure for the results section in the Abstract, and the first paragraph (principal findings) in the the Discussion section.

Response: We have revised as follows:

• We have consistently used implementation and not utilization as it better reflects the message within the manuscript.

• We have restructured the results section and the discussion sections. We have presented the results under 5 themes instead of the 6 as suggested. We could not find enough quotes to create a stand-alone quote on “Parents Fear of Bubble CPAP” The five themes are as follows:

o Inadequate training of healthcare providers in preparation for use, 

o Rigid division of roles and responsibilities, 

o Lack of effective communication, 

o Human resource constraints and 

o Lack of equipment and infrastructure.

These five themes form the basis of our results presentations and are on lines 209-390

Line 186: there are inconsistencies in the order of presenting broad categories (health system, providers, caregivers) throughout the manuscript.

Response: Secondary to the revision of the presentation of our results, this inconsistency has been cleared off.

Abstract Results

Line 51: There are personal, patient and health system level factors Line 186: at the health system, provider and caregiver level

Response: This has been corrected to and reads as: There are provider, caregiver and health system level factors that influence the implementation of bubble CPAP among neonates in Malawian health facilities. Refer to lines 49-50.

Lines 195 percentages are usually not used in a qualitative paper.

Response: We have removed percentages 

Line 198 provide more details about what monitoring means

Response: We have provided more details on monitoring. We have included the following:

Monitoring included ensuring that the connections to the machines remained intact and also the checking the baby’s vital signs at specified intervals as in line 225.

Discussion

The discussion should start with main findings.

Response: Our discussion now starts with the summary of the main findings as follows: 

Factors affecting bubble CPAP implementation were interconnected and the barriers included inadequate healthcare provider training in preparation for use, rigid division of roles and responsibilities among healthcare providers, lack of effective communication, human resource constraints, and inadequate equipment and infrastructure. Factors that facilitated implementation of bubble CPAP were comprehensive training, team cohesion, dedicated nurses for the nursery and consistent availability of electricity and equipment. Refer to lines 393-398.

There should be a “strength and limitations” section in the discussion. P14, Lines 358-360 is a strength of the study

Response: We have included a section on “strengths and limitations” after the discussion. 

Our study adds to the existing literature by closely examining the factors that influence implementation of bubble CPAP from healthcare professionals’ perspectives. Furthermore our paper navigates through the process of bubble CPAP implementation from training, decision-making to initiate, putting the baby on bubble CPAP, monitoring and weaning. Finally, preterm neonates often have complex medical needs that are not limited to respiratory distress. Comprehensive newborn care is therefore required to achieve tangible improvements in neonatal mortality. While we did not explore other neonatal care interventions in our bubble CPAP interviews, we are cognizant that CPAP should be a component of a larger package of newborn care. 

Our study was conducted in hospitals with varying authority structures, ownership, and across different cadres of healthcare providers, therefore offering a holistic view of the implementation of bubble CPAP. Although our study followed a qualitative approach, which may limit generalisations of the results, our findings offer concepts that can be taken onboard when implementing bubble CPAP in other hospitals. Refer to lines 492-506.

You should present the themes in the same order than in the Abstract and the Results section

Response: We have revised the presentation of themes in the discussion section, it now follows the flow of the results section.

The following points in the Discussion are not presented beforehand in the Results section:

• Line 429-430 “However, health professionals in our study rarely discussed the need for nasal saline drops as a challenge”.

• Line 442-44 “To avoid the increased burden of monitoring a baby on bubble CPAP, some nurses shared that they left the baby on nasal oxygen which reportedly required less issues to monitor.”

Suggestions: either present them in the results section, or remove them.

Response: We have removed the sections as they were poorly substantiated.

Reviewer #2: This was a very well-written manuscript from a study that was well-designed and conducted. The details of the study are clearly stated, and justification of methods are clearly outlined. The data analysis was rigorously done. The tables and figures are well-presented and described in the text.

Response: Thank you for the comment, we appreciate.

Discussion:

Line 395 “Bubble” remove capital.

Response: We have revised it to bubble.

Although alluded to in the last paragraph of this section, it would be useful for the authors to provide more details on their reflections of the limitations of this study and how they have attempted to mitigate them.

Response: We have added a section of strengths and limitation after the discussion 

Our study adds to the existing literature by closely examining the factors that influence implementation of bubble CPAP from healthcare professionals’ perspectives. Furthermore our paper navigates through the process of bubble CPAP implementation from training, decision-making to initiate, putting the baby on bubble CPAP, monitoring and weaning. Finally, preterm neonates often have complex medical needs that are not limited to respiratory distress. Comprehensive newborn care is therefore required to achieve tangible improvements in neonatal mortality. While we did not explore other neonatal care interventions in our bubble CPAP interviews, we are cognizant that CPAP should be a component of a larger package of newborn care. 

Our study was conducted in hospitals with varying authority structures, ownership, and across different cadres of healthcare providers, therefore offering a holistic view of the implementation of bubble CPAP. Although our study followed a qualitative approach, which may limit generalisations of the results, our findings offer concepts that can be taken onboard when implementing bubble CPAP in other hospitals. Refer to lines 493- 506.

Well done to the team.

Response: Thank you and appreciated.

Do not hesitate to contact me should you have any questions or areas that need clarification.

Yours Faithfully,

Alinane Linda Nyondo-Mipando

---

## [Decision Letter · Decision Letter 1]

22 Nov 2019

PONE-D-19-22572R1

Barriers and enablers of implementing bubble Continuous Positive Airway Pressure (CPAP): Perspectives of health professionals in Malawi

PLOS ONE

Dear Dr Nyondo-Mipando,

Thank you for submitting your manuscript to PLOS ONE. After careful consideration, we feel that it has merit but does not fully meet PLOS ONE’s publication criteria as it currently stands. Therefore, we invite you to submit a revised version of the manuscript that addresses the points raised during the review process.

There are a few minor comments to be addressed.

We would appreciate receiving your revised manuscript by Jan 06 2020 11:59PM. To enhance the reproducibility of your results, we recommend that if applicable you deposit your laboratory protocols in protocols.io, where a protocol can be assigned its own identifier (DOI) such that it can be cited independently in the future. For instructions see: http://journals.plos.org/plosone/s/submission-guidelines#loc-laboratory-protocols

We look forward to receiving your revised manuscript.

Kind regards,

Charles A. Ameh, PhD, MPH, FWACS (OBGYN), FRCOG

Academic Editor

PLOS ONE

Additional Editor Comments (if provided):

Thanks for addressing most of the comments, there are a few minor comments to be addressed before your manuscript can be accepted.

Reviewers' comments:

Reviewer's Responses to Questions

**Comments to the Author**

1. If the authors have adequately addressed your comments raised in a previous round of review and you feel that this manuscript is now acceptable for publication, you may indicate that here to bypass the “Comments to the Author” section, enter your conflict of interest statement in the “Confidential to Editor” section, and submit your "Accept" recommendation.

Reviewer #1: (No Response)

Reviewer #2: All comments have been addressed

2. Is the manuscript technically sound, and do the data support the conclusions?

Reviewer #1: Yes

Reviewer #2: Yes

3. Has the statistical analysis been performed appropriately and rigorously? 

Reviewer #1: N/A

Reviewer #2: N/A

4. Have the authors made all data underlying the findings in their manuscript fully available?

Reviewer #1: No

Reviewer #2: No

5. Is the manuscript presented in an intelligible fashion and written in standard English?

Reviewer #1: Yes

Reviewer #2: Yes

6. Review Comments to the Author

Reviewer #1: Thank you to the authors for making the amendments. I think the paper is well written and structured, and conclusions are supported by the data. I fully support the publication of this paper, which I think fulfill the Plos One criteria for publication.

I have few minor final comments that should be addressed:

Results section:

I think the Participant characteristics and Factors that influence implementation of bubble CPAP subtitles are not useful. I suggest removing those, and keeping subtitles only for the 5 main themes that you have found:

O Inadequate training of healthcare providers in preparation for use,

O Rigid division of roles and responsibilities,

O Lack of effective communication,

O Human resource constraints and

O Lack of equipment and infrastructure

This will allow readers easily identifying your main results and understanding the structure of your findings.

Line 191-192: I think the median age of participants has probably no impact on how participants responded to your research question. I suggest removing it in the text and table, and rather adding median years of CPAP use in the table; this is important because it allows knowing whether participants have a substantial experience of using CPAP, which probably had an impact on what they said in the interviews.

Line 206: Perhaps add here that there are 5 types of barriers. This will help first time readers to grasp quickly the structure of the results section; something like: Five main types of barriers to the implementation/use(?) were identified and were: …..

Discussion section:

Lines 496-501 are not really limitations of your study, they are findings/recommendations.

The team should reflect about whether the findings should be interpreted in light of some limitations. For example:

• Could the way of selecting participants have influenced what participants said? Who was the member of the research team (line 143) who initially contacted potential participants? Was he/she in a position of authority? were there potential hierarchical/power issues that could have influenced the responses?

• The five interviewers had different backgrounds, could that have influenced what participants said? If yes, is this an issue?

• Were nurses as fluent in English as doctors? Given that interviews were in English, would that have influenced what nurses were able to share?

• Was it clear to participants that interviewers were (presumably?) hired by IMCHA? Would that have influenced what they said about an intervention that presumably IMCHA is keen to implement?

• Although your objective was clearly to explore the perceptions of healthcare workers that are important in terms of implementation of CPAP, perhaps you should acknowledge that the influence of healthcare workers on the implementation decisions is relatively limited; in other words, there are probably other (more important?) barriers at higher levels in the health system; perhaps you could suggest that other studies with other cadres of staff with more influence should complement your study?

Reviewer #2: The authors have addressed the concerns that I raised comprehensively. I would be happy for this manuscript to be published.

7. PLOS authors have the option to publish the peer review history of their article (what does this mean?). If published, this will include your full peer review and any attached files.

Reviewer #1: Yes: Juan Emmanuel Dewez

Reviewer #2: No

---

## [Author Response · Author response to Decision Letter 1]

15 Dec 2019

Dear Editor,

On behalf of the authors, I would like to thank the reviewers for their constructive review which has improved the manuscript. Please find below responses to the comments raised in the last review:

Reviewer #1: Thank you to the authors for making the amendments. I think the paper is well written and structured, and conclusions are supported by the data. I fully support the publication of this paper, which I think fulfill the Plos One criteria for publication.

Response: The authors appreciate this comment and sincerely thank you for thorough, constructive, open feedback.

I have few minor final comments that should be addressed:

Results section:

I think the Participant characteristics and Factors that influence implementation of bubble CPAP subtitles are not useful. I suggest removing those, and keeping subtitles only for the 5 main themes that you have found:

O Inadequate training of healthcare providers in preparation for use,

O Rigid division of roles and responsibilities,

O Lack of effective communication,

O Human resource constraints and

O Lack of equipment and infrastructure

This will allow readers easily identifying your main results and understanding the structure of your findings.

Response: the two subtitles have been deleted as suggested.

Line 191-192: I think the median age of participants has probably no impact on how participants responded to your research question. I suggest removing it in the text and table, and rather adding median years of CPAP use in the table; this is important because it allows knowing whether participants have a substantial experience of using CPAP, which probably had an impact on what they said in the interviews.

Response: the reference to age has been delete. We have the overall median years of CPAP use in the narrative before the table 

Line 206: Perhaps add here that there are 5 types of barriers. This will help first time readers to grasp quickly the structure of the results section; something like: Five main types of barriers to the implementation/use(?) were identified and were: …..

Response: We specified that there were “Five main types of barriers” identified.

Discussion section:

Lines 496-501 are not really limitations of your study, they are findings/recommendations.

Response: Acknowledged. We elaborated on study limitations in the strengths/limitations section. Please see below. 

The team should reflect about whether the findings should be interpreted in light of some limitations. For example:

• Could the way of selecting participants have influenced what participants said? Who was the member of the research team (line 143) who initially contacted potential participants? Was he/she in a position of authority? were there potential hierarchical/power issues that could have influenced the responses?

• The five interviewers had different backgrounds, could that have influenced what participants said? If yes, is this an issue?

• Were nurses as fluent in English as doctors? Given that interviews were in English, would that have influenced what nurses were able to share?

• Was it clear to participants that interviewers were (presumably?) hired by IMCHA? Would that have influenced what they said about an intervention that presumably IMCHA is keen to implement?

Response: We have reflected and added information as follows:

Recruitment and Selection (Line 143-152)

Participants were approached face-to-face or by phone by a members of the research team at both the tertiary and district hospitals and introduced themselves as IMCHA study team members. At the tertiary hospital a registrar approached the possible participants and later the research assistants followed with making appointment with the potential candidate for informed consent procedures and data collection. The registrar did not conduct any interviews. At the district level a program manager who introduced himself as part of the study and was outside the medical hierarchy, connected with participants to book appointments with the potential participants and the research assistants also followed with informed consent procedures and data collection.

We have revised the strength and limitation sections as follows to better clarify the issues raised:

Our study adds to the existing literature by closely examining the factors that influence implementation of bubble CPAP from healthcare professionals’ perspectives. Furthermore, our paper navigates through the process of bubble CPAP implementation from training, decision-making to initiate, putting the baby on bubble CPAP, monitoring and weaning. Although our study followed a qualitative approach, which may limit generalisations of the results, our findings offer concepts that can be taken onboard when implementing bubble CPAP in other hospitals. It is important to emphasize that preterm neonates often have complex medical needs that are not limited to respiratory distress. Comprehensive newborn care is therefore required to achieve tangible improvements in neonatal mortality. While we did not explore other neonatal care interventions in our bubble CPAP interviews, we are cognizant that CPAP should be a component of a larger package of newborn care. 

Researchers’ background and motivations inevitably influence the researcher-study participant dynamics and therefore participants’ responses. Healthcare providers participating in our study were aware that all the researchers were part of the IMCHA program, which supports implementation of neonatal technologies, and works in close partnership with RICE Institute, which supports delivery and utilization of bubble CPAP in Malawi. While the researchers made it clear that the objective of this research study was not to evaluate healthcare providers’ performance or to provide additional hospital resources, we acknowledge that respondent’s desirability bias could not be avoided. A similar study conducted by an independent research team may have elicited additional barriers to bubble CPAP implementation that were omitted during our interviews.

Researchers conducting the interviews came from clinical (nurses) and non-clinical (public health) backgrounds. Upon reflecting on the interviews, nurses noted that they had to reemphasize their role as researchers rather than clinicians with study participants and ask them to explain concepts that may only be familiar to nurses in lay language. 

Finally, our study focused on the perspectives and experiences of frontline healthcare providers and district level health management. These findings will have to be triangulated with perspectives from healthcare policy decision makers, such as government officials, who are able to speak about barriers and facilitators to bubble CPAP implementation from a system level perspective. 

In lines 178-180, we have clarified the use of English language across the different professions in Malawi as follows:

Interviews were conducted mainly in English, which is the language of instruction for health care professionals in Malawi, though participants were invited to use the local language of Chichewa if they were more comfortable doing so.

• Although your objective was clearly to explore the perceptions of healthcare workers that are important in terms of implementation of CPAP, perhaps you should acknowledge that the influence of healthcare workers on the implementation decisions is relatively limited; in other words, there are probably other (more important?) barriers at higher levels in the health system; perhaps you could suggest that other studies with other cadres of staff with more influence should complement your study?

Response: We have included the following in lines 491-493

Future research should also consider inclusion of decision makers and other cadres of staff with more influence on what interventions gets to be implemented within the Malawi Health System. 

We have also included the following in the conclusion section, lines 540-542

Our study was conducted in hospitals with varying authority structures, ownership, and across different cadres of healthcare providers, therefore offering a holistic view of the implementation of bubble CPAP. 

Reviewer #2: The authors have addressed the concerns that I raised comprehensively. I would be happy for this manuscript to be published.

Response: We appreciate for taking the time to review our manuscript.

Yours Faithfully,

Alinane Linda Nyondo-Mipando (Corresponding Author)

---

## [Editor Report · Decision Letter 2]

28 Jan 2020

Barriers and enablers of implementing bubble Continuous Positive Airway Pressure (CPAP): Perspectives of health professionals in Malawi

PONE-D-19-22572R2

Dear Dr. Nyondo-Mipando,

We are pleased to inform you that your manuscript has been judged scientifically suitable for publication and will be formally accepted for publication once it complies with all outstanding technical requirements.

With kind regards,

Charles A. Ameh, PhD, MPH, FWACS (OBGYN), FRCOG

Academic Editor

PLOS ONE
---

## [Editor Report · Acceptance letter]

3 Feb 2020

PONE-D-19-22572R2 

Barriers and enablers of implementing bubble Continuous Positive Airway Pressure (CPAP): Perspectives of health professionals in Malawi 

Dear Dr. Nyondo-Mipando:

I am pleased to inform you that your manuscript has been deemed suitable for publication in PLOS ONE. Congratulations! Your manuscript is now with our production department. 

With kind regards,

on behalf of

Dr. Charles A. Ameh 

Academic Editor

PLOS ONE